# MUG CCArly: A Novel Autologous 3D Cholangiocarcinoma Model Presents an Increased Angiogenic Potential

**DOI:** 10.3390/cancers15061757

**Published:** 2023-03-14

**Authors:** Silke Schrom, Florian Kleinegger, Ines Anders, Thomas Hebesberger, Christina Karner, Laura Liesinger, Ruth Birner-Gruenberger, Wilfried Renner, Martin Pichler, Regina Grillari, Ariane Aigelsreiter, Beate Rinner

**Affiliations:** 1Division of Biomedical Research, Medical University of Graz, 8036 Graz, Austria; 2Diagnostic and Research Institute of Pathology, Medical University of Graz, 8036 Graz, Austria; 3BiotechMed-Graz, 8010 Graz, Austria; 4Institute of Chemical Technologies and Analytics, Faculty of Technical Chemistry, Technische Universität Wien, 1060 Vienna, Austria; 5Clinical Institute of Medical and Chemical Laboratory Diagnostics, Medical University of Graz, 8036 Graz, Austria; 6Translational Oncology Department, University Hospital of Augsburg, 86156 Augsburg, Germany; 7Evercyte GmbH, 1190 Vienna, Austria; 8Institute of Bioprocess Science and Engineering, University of Natural Resources and Life Sciences—BOKU Wien, 1180 Vienna, Austria

**Keywords:** co-culture tumor model, spheroid, tumor microenvironment, cancer-associated fibroblasts, proteomics

## Abstract

**Simple Summary:**

Cholangiocarcinoma is a rare, aggressive, and heterogeneous malignancy of the bile duct. A typical feature of cholangiocarcinoma is that the cancer cells are embedded in a dense stroma. However, the functional role of the reactive tumor stroma has not been fully elucidated, which is certainly due to the lack of suitable in vitro models. Tumor stromal cells or tumor-associated fibroblasts play a central role in angiogenesis, metastasis, and the development of resistance to therapy in cholangiocarcinoma. We successfully isolated tumor cells and tumor-associated fibroblasts from a patient, generated cell lines, characterized the cells in detail, and established an innovative autologous tumor model. By detailed characterization of the tumor/tumor-associated fibroblast model we demonstrated that tumor cells interact with tumor-associated fibroblasts. This model can be used to study tumor stromal crosstalk, tumor angiogenesis and invasion, and the development of drug resistance.

**Abstract:**

Cholangiocarcinoma (CCA) are characterized by their desmoplastic and hypervascularized tumor microenvironment (TME), which is mainly composed of tumor cells and cancer-associated fibroblasts (CAFs). CAFs play a pivotal role in general and CCA tumor progression, angiogenesis, metastasis, and the development of treatment resistance. To our knowledge, no continuous human in vivo-like co-culture model is available for research. Therefore, we aimed to establish a new model system (called MUG CCArly) that mimics the desmoplastic microenvironment typically seen in CCA. Proteomic data comparing the new CCA tumor cell line with our co-culture tumor model (CCTM) indicated a higher gene expression correlation of the CCTM with physiological CCA characteristics. A pro-angiogenic TME that is typically observed in CCA could also be better simulated in the CCTM group. Further analysis of secreted proteins revealed CAFs to be the main source of these angiogenic factors. Our CCTM MUG CCArly represents a new, reproducible, and easy-to-handle 3D CCA model for preclinical studies focusing on CCA-stromal crosstalk, tumor angiogenesis, and invasion, as well as the immunosuppressive microenvironment and the involvement of CAFs in the way that drug resistance develops.

## 1. Introduction

Cholangiocarcinoma (CCA), a globally frequently diagnosed gastro-intestinal cancer type, develops in the cells of the bile ducts and comprises three anatomical subtypes: intrahepatic CCA (iCCA) occurs in the left or right hepatic duct, perihilar CCA (pCCA) in the bifurcation of both main ducts, and distal CCA (dCCA) in the extrahepatic common bile duct [1]. The origin and location of tumor cells influence their growth behavior, molecular pathways, marker expression, and metastasis formation, and thus impact on survival and even molecular stratified treatment strategies [2,3].

Although most CCA cases are pCCA, very few respective cell lines are available, which leaves a void in preclinical models for this specific form of CCA, which is also called a Klatskin tumor [4].

As a rare cancer in developed countries, CCA affects 0.3–6 in 100,000 individuals each year, but accounts for 2% of cancer-related deaths with a 5-year survival rate of 7–20% [5,6]. CCA often metastasizes at early stages, remains asymptomatic, and is usually diagnosed at advanced stages. Furthermore, CCA is resistant to conventional cytotoxic therapy, leaving resection as the only curative treatment option, although high recurrence rates have been reported after surgical procedures [7]. If, as in most cases, resection is not applicable, palliative cisplatin-based treatment is chosen [8].

Considering typical hallmarks, CCA is characterized by its highly inflammatory, desmoplastic, and hyper-vascularized tumor microenvironment (TME), which mainly consists of CAFs. These stromal cells release a multitude of pro-inflammatory factors and reactive extracellular matrix components, thereby contributing to therapy resistance, angiogenesis, tumor progression, and the invasion of cancer cells [9,10,11,12]. Glycoproteins, like tenascin C (TNC) and fibronectin 1 (FN1), as well as glycosaminoglycans, aminoglycans, proteoglycans, and fibrous components (such as collagen), shape the TME. The dense TME is predominantly modified by CAFs and influences drug penetration. Additionally, angiogenesis and consequently tumor cell migration depend on ECM remodeling and are associated with lower survival rates in CCA [12,13,14].

To mimic this tumor microenvironment as close as possible, we chose the 3D multicellular spheroid culture method. Three-dimensional multicellular models simulate the in vivo pathophysiology of solid tumors more closely in contrast to conventional 2D cultures. This was proven by an increased cellular heterogeneity upon the 3D culture of breast cancer and melanoma cells [15,16]. In 3D spheroid cultures, the dense cell-to-cell interaction and gradients such as nutrients, metabolic products, pH, and gases, affect gene expression, cell-to-cell signaling, and drug resistance. These gradients can simulate hypoxia, which is typical in solid tumors, favoring an angiogenic microenvironment. Co-culturing tumor cells in the presence of stromal cells promotes this proangiogenic microenvironment further and drive cellular complexity and plasticity. The addition of CAFs to tumor spheroid cultures influences tumor progression and drug resistance formation through an increased production of ECM components. Such co-cultured spheroids require no ECM supplementation. Therapeutic approaches already take the pro-tumorigenic TME into account [17,18], but are still in the early stages of development.

With this background and to our knowledge that no ready-to-use CCA 3D in vitro multicellular model exist, we aimed to develop an in vivo-like 3D co-culture tumor model (CCTM) for which CCA tumor cells and the corresponding immortalized CAFs from a Klatskin tumor had been isolated. Both cell lines and the 3D model system were comprehensively characterized. Focusing on angiogenesis, CCA single cultures and the CCTM were then compared to well-known physiological CCA characteristics. This new CCA in vitro model can easily be reproduced under standard laboratory conditions and could be used for basic research in CCA and for preclinical studies.

## 2. Materials and Methods

### 2.1. Patient History

A 72-year-old, female Caucasian patient was hospitalized and diagnosed with CCA. First tests demonstrated elevated carbohydrate antigen 19.9 (CA19.9) levels. The patient underwent a hemihepatectomy along with a resection of the *Lobus caudatus*. One part of the tissue specimen was used for diagnostic purposes and another small piece was placed into fresh cell culture media for research. The clinical report confirmed a Bismuth–Corlette type-IV Klatskin tumor of the extrahepatic bile ducts (G2, pT4, N2, Pn1). As the tumor could not be resected in sano, adjuvant chemotherapy was considered but not implemented due to postoperative complications. In addition, the patient’s health deteriorated steadily, and she was discharged to palliative care. The patient gave written informed consent for the study specific procedure. All experiments were conducted and approved according to the guidelines of the ethics committee of the Medical University of Graz (vote #28–294ex15/016).

### 2.2. Cell Culture

A tumor piece of approximately 6 mm^3^ was cut into 1–2 mm^3^ sized pieces during 10× Penicillin/Streptomycin incubation (Gibco, Life Technologies, Darmstadt, Germany) and taken into culture after Accutase (Biozym, Hessisch Oldendorf, Germany) digestion for 30 min at room temperature (RT). Pieces were incubated under humidified conditions at 37 °C, 5% CO_2_. The cultivation of primary cells is a challenge. Due to the small tumor size and the heterogeneity of the specimen, growth was initially very slow. It was only after a month that tumor cells could be separated from stromal cells using different detachment time points. Spindle-shaped stromal cells could be detached after less than one minute of proteolytic and collagenolytic dissociation using Accutase, whereas tumor cells detached from plastic surface after three to five minutes. Tissue pieces and both cell lines were cultured in DMEM/F12 (Gibco, Life Technologies, Darmstadt, Germany) supplemented with 10% FBS (M&B Stricker, Bernried, Germany), 2 mM L-Glutamine (Gibco, Life Technologies, Darmstadt, Germany), and 1× Pen/Strep (Gibco, Life Technologies, Darmstadt, Germany). Tumor cells were named MUG CCArly. Tumor-surrounding CAFs were immortalized in cooperation with the company Evercyte using human telomerase reverse transcriptase (hTERT) system named CCArly CAF. A549 control cells were cultivated in DMEM (Gibco, Life Technologies, Darmstadt, Germany) and were supplemented with 10% FBS, 2mM L-Glutamine, and 1× P/S.

Both established cell lines were consistently checked for mycoplasma by PCR (Minerva Biolabs, Berlin, Germany). Short tandem repeat (STR) profiling (Promega, Madison, WI, USA) was used for cell authentication. All cell lines were used for experiments below passage 30 and for CAF below passage 20. Microscopic pictures were taken on an Eclipse Ti2 inverted microscope (Nikon, Tokyo, Japan), with a numerical aperture of 0.30, and a DS-Fi2 camera (Nikon, Tokyo, Japan) at RT. Analysis was performed with the NIS-Elements BR 5.02.00 software (Nikon, Tokyo, Japan).

### 2.3. 3D Cell Culture

For 3D cultures, both cell lines, CCArly CAF and MUG CCArly, were harvested from 2D cultures. Single suspensions of these cell lines alone and in combination were then seeded in ultra-low attachment 96-well-plates (Corning, NY, USA) at a cell density of 5000 cells/well in 100 μL media/well to build 3D spheroid cultures. MUG CCArly single-culture spheroids were supplemented with 28 μg/mL bovine collagen I (Gibco, Life Technologies, Darmstadt, Germany) for stable spheroid formation. Half media changes were performed every second to third day. For immunocytochemistry staining, spheroids were harvested on day six, fixed in 4% formalin, and embedded in paraffin (FFPE). Supernatants were collected on cultivation days three and seven for Luminex xMAP technology. In total, 50 μL per well and condition was pooled for 20 biological replicates, and stored at −80 °C until further use.

### 2.4. Immunohistochemistry

FFPE samples of the primary tumor and 3D spheroids, as well as cell lines grown on chamber slides and fixed with 4% formalin, were stained at the Diagnostic and Research Institute of Pathology of the Medical University of Graz. All markers are listed in Appendix A.

### 2.5. Cell Line Identification by Short Tandem Repeat (STR) Analysis

DNA was isolated and co-amplificated using the PowerPlex^®^ 16 HS System kit (Promega Corporation, Madison, WI, USA), fractioned on a 3730 DNA analyzer (Applied Biosystems, MA, USA), and data were evaluated using the Genemapper software 3.7 (Applied Biosystems, Waltham, MA, USA).

### 2.6. Telomerase Activity

A telomerase-repeated amplification protocol (TRAP) assay was conducted to verify hTERT immortalization of CCArly CAF. A549 was used as a positive control cell line for telomerase activity according to Ludlow et al. [19]. Both established cell lines and A549 cells were measured in biological triplicates. Briefly, cell pellets of 2.5 × 10^5^ cells were snap frozen for later use. The TRAP assay was done according to protocol [20]. Relative telomerase concentration was normalized to A549 control cells.

### 2.7. Tumorigenicity

Four-to-five-week-old CR ATH HO ((Crl:NU(NCr))-Foxn1^nu^ mice (Charles River Laboratories, Kent, UK) were housed under conventional conditions in ventilated cages and fed ad libitum. After an acclimation period of one week, mice were divided into three groups (*n* = 5 per group): group A: MUG CCArly, group B: CCArly CAF, and group C: 1 + 4 MUG CCArly + CCArly CAF, respectively. All mice were injected with a 27G needle and a total of 1 × 10^6^ cells in 100 μL PBS subcutaneously (s.c.) into the right posterior flank. Inoculation of cells was monitored once a week by high-frequency ultrasound imaging (HF-US). Cell injection and HF-US were performed under anesthesia with a constant administration of 2% isoflurane in a permanent airflow of two liters per minute. Nine weeks post-injection, animals were sacrificed. Afterwards, histopathological examination was performed. Tissues were fixed in 4% paraformaldehyde solution for 24 h and embedded in paraffin. All of the animal work complies with the ARRIVE guidelines and has been approved by the committee for institutional animal care and use at the Austrian Federal Ministry of Science and Research (BMWFW) (vote 66.010/0046-WF/V/3b/2016).

### 2.8. Ultrasound Imaging

HF-US was performed for tumor monitoring using a Vevo3100 HF-US system (Fujifilm VisualSonics, Inc., Toronto, ON, Canada) with a 40 MHz (MX700) transducer (Fujifilm VisualSonics). Sagittal and transversal images of the region of interest were obtained.

### 2.9. Protein Isolation of 3D Cultures

Spheroids were harvested on day three and seven of cultivation for proteomic analysis, and washed with DPBS (Gibco, Life Technologies, Darmstadt, Germany) before cell lysis in 130 μL 1X RIPA (99%)/protease-inhibitor cocktail (PIC) (1%) (Sigma-Aldrich, St. Louis, MO, USA). To ensure efficient cell lysis, precipitates were pipetted vigorously and vortexed for three minutes. Lysates were then stored at −80 °C before protein quantification. Spheroid lysates of all five experiments were thawed and forwarded for a detergent compatible (DC) assay according to the manufacturer’s protocol (Bio-Rad, Hercules, CA, USA).

### 2.10. Liquid Chromatography–Tandem Mass Spectrometry (LC-MS/MS)

Proteins were precipitated with 10 mM NaCl in four volumes of acetone for 10 min at RT. Solubilization was performed in 50% 2,2,2-Trifluoroethanol (TFE) in 50 mM Tris-HCl (pH 8.5). A total of 10 μg each was then reduced by 10 mM Tris(2-carboxyethyl)phosphine (TCEP) and alkylated with 40 mM chloroacetamide for 10 min at 95 °C. A total of 50 mM ammonium bicarbonate was used to dilute samples to less than 10% TFE. Protein digestion occurred over night in Promega Trypsin/LysC Mix (25:1) at 37 °C, at 550 rounds per minute. Trifluoracetic acid was used for acidification of 4 μg each to a final concentration of 1%. Samples were finally desalted by Styrene Divinylbenzene-Reversed Phase Sulfonate (SDB-RPS) stage-tips. Peptide extracts were separated with the UltiMate 3000 RSLCnano Dionex system (Thermo Fisher Scientific, Waltham, MA, USA) using an Ionopticks Aurora Series UHPLC C18 column (250 mm × 75 μm, 1.5 μm) (Ionopticks, Fitzroy, Australia) by applying an 86.5 min gradient at a flow rate of 400 nLmin at 40 °C (solvent A: 0.1% formic acid in water; solvent B: acetonitrile with 0.1% formic acid; 0–5.5 min: 2% B; 5.5–25.5 min: 2–10% B; 25.5–45.5 min: 10–25% B, 45.5–55.5 min: 25–37% B, 55.5–65.5 min: 37–80% B, 65.5–75.5 min: 80% B; 75.5–76.5 min: 80–2% B; 76.5–86.5 min: 2% B). The timsTOF Pro mass spectrometer (Bruker Daltonics, Bremen, Germany) was operated as follows: positive mode, enabled trapped ion mobility spectrometry (TIMS), 100% duty cycle (ramp 100 ms); source capillary voltage (V): 1500 V; dry gas flow: 3 L/min, 180 °C; scan mode: data-independent parallel accumulation–serial fragmentation (diaPASEF), previously described by Meier [21], using 21 × 25 Th isolation windows, *m*/*z* 475–1000; 0 Th overlap between windows. Two and three isolation windows were fragmented per TIMS ramp after the MS1 scan, respectively (overall DIA cycle time: 1 s).

### 2.11. Protein Data Evaluation

Protein data were processed with DIA-NN (version 1.8) [22,23]. The SwissProt homo sapiens database (downloaded on 17 December 2020 with 20,461 sequences including common contaminants) was used to interpret data with a false discovery rate (FDR) cutoff at 1%. Settings were set as follows: deep learning-based spectra: enabled; retention time prediction: enabled; N-terminal methionine excision: enabled; cysteine carbamidomethylation: fixed; methionine oxidation: variable; trypsin missed cleavages: max. 2; minimum fragment *m*/*z*: 200, max fragment *m*/*z*: 1800; minimum peptide length: 7 amino acids (AA); and max peptide length: 30 AA. Mass accuracy was automatically optimized by DIA-NN based on the first run of the experiment.

The mass spectrometry proteomics datasets (including the DIA-NN version used to process the data) have been deposited to the ProteomeXchange Consortium via the PRIDE partner repository [24] with the dataset identifier PXD037649.

Using the Perseus software version 1.6.15.0 [25], MS protein group quantities were further analyzed. Briefly, intensities were log2 transformed to lower outlier effects and filtered for four valid values per group, replacing missing values randomly from the Gaussian distribution (width: 0.3, downshift: 1.8). The string database (www.string-db.og, accessed on 17 March 2022) was used to display protein interactions [26].

### 2.12. Luminex xMAP Technology

Concentrations of secreted factors were measured using the human magnetic Luminex Assay System according to the manufacturer’s protocol. A Bio-Plex 200 multiplex suspension array system (Bio-Rad, Hercules, CA, USA) was used for signal measurements and detection was analyzed using the Bio-Plex 5.0 Software (Bio-Rad, Hercules, CA, USA). Supernatants of spheroids were taken on day three and seven for subsequent proteomic analysis. Pools of 20 wells each were stored at −80 °C until further use. All samples were measured in duplicates and three biological replicates.

### 2.13. Statistical Analysis

To identify altered protein groups for the proteomic analysis, two-sample student’s t-tests and multiple correction testing applying a permutation-based FDR approach were performed. To compare DNA indices of both new cell lines obtained by cell cycle analysis, as well as for protein concentrations and protein expressions comparing both 3D single cultures to the CCTM group, two-tailed student’s *t*-tests were conducted using GraphPad Prism 9.3.1 (GraphPad Software, San Diego, CA, USA).

## 3. Results

### 3.1. Establishment and Characterization of the New CCA Cell Line MUG CCArly and Its Autologous hTERT Immortalized CCArly CAF

A primary Klatskin tumor tissue originating from a patient without prior therapy was taken into culture. Stromal spindle-shaped cells appeared after seven days, and single tumor cells appeared after two weeks (Figure 1A).

The tissue specimen, toT establish the cell lines was very small and only a few tumor cells could be detected after two weeks with a slow growth behavior. Despite the effort to separate both cell types from one another, stromal cells overgrew the tumor cells quickly. Stromal cells were separated several times from the primary culture, whereas slowly growing primary tumor cells were left adhered to the cell culture surface. A pure and fast growing CCA cell line could therefore be established after eight months.

This CCA cultures were visible as cobblestone-shaped monolayers, accumulating in colonies with a splitting rate of 1:6–1:10 (Figure 1B). This new CCA cell line was termed MUG CCArly. Meanwhile, primary cells at passage three were sent to the company Evercyte for hTERT immortalization of stromal CAFs. CCArly CAFs were split at an 80% confluency at a rate of 1:2 (Figure 1C). Both new cell lines were subcultivated over 60 passages to additionally indicate their immortal character.

IHC staining of both cell lines and the primary tumor tissue confirmed a stable expression of CCA tumor markers CK7, CK19, and e-cadherin, and the positive staining of stromal cells for vimentin and CAF markers α-SMA, TNC, and PDGFRB (Figure 2 and Appendix A).

To define the DNA ploidy of both new cell lines, MUG CCArly and CCArly CAF, a cell cycle analysis was performed, including peripheral blood mononuclear cells (PBMCs) as a standard (G0) for diploid cells (Appendix A). DNA ploidy (G0/G1), based on the PBMC control cells, resulted in a hyperdiploidy for MUG CCArly cells with a ratio of 1.73 ± 0.07. According to Ross et al., CCArly CAF could be classified as diploid with a G0/G1 ratio of 1.13 ± 0.04 (Appendix A) [27].

To confirm hTERT immortalization of CCArly CAF, a TRAP assay was performed including MUG CCArly cells. After normalization to the control cell line A549, MUG CCArly presented a relative telomerase activity of 206.66 ± 78.35% and CCArly CAF represented an even higher activity at 267.69 ± 85.84% (Appendix A).

STR authentication of both cell lines matched the primary tumor tissues throughout the experiments with two changes for MUG CCArly on locus D7S820 and Penta D (Appendix A).

### 3.2. In Vivo Studies Confirm the Tumorigenic Potential of MUG CCArly

To verify the tumorigenic potential of MUG CCArly cells and non-tumorigenic behavior of CCArly CAF and CR ATH HO nude mice were injected s.c. with both cell lines alone and in a 1 + 4 combination, with the majority constituting CCArly CAF. Five weeks post-injection, first xenograft growth could be observed for mice injected with MUG CCArly, as well as the 1 + 4 combination group (Figure 3A). Until sacrification at nine weeks, no tumor growth was monitored for the CCArly CAF group. Resected tumors of MUG CCArly were bigger than xenografts of the combination group (Figure 3B). For the IHC staining, no differences were observed between both groups, as CCA cells stained positive for CK7, CK19, and e-cadherin, and exhibited a proliferative state marked by Ki-67 positive cells (Figure 3C). Vimentin, which stained specifically human stromal cells, was not detected in the combination group, pointing out a degradation of CCArly CAF in vivo. The growth of a representative xenograft for each group was detected by HF-US (Figure 3D).

### 3.3. Quality Assessment of the 3D In Vitro Co-Culture Tumor Model

Three-dimensional spheroid cultures, especially under co-culturing conditions, were characterized prior to additional 3D analyses. First, the optimal spheroid development over time was ensured by seeding MUG CCArly and CCArly CAF in ULA plates separately, and in a 1 + 4 ratio (CCTM), respectively. CCArly CAF and the CCTM both formed stable circular spheroids within 24 h (Figure 4A and Appendix A). MUG CCArly cells alone remained as single cells, even after incubation periods over five days (Appendix A). The medium was supplemented with collagen I to promote a stable spheroid formation for MUG CCArly (Figure 4A). The resulting MUG CCArly spheroids resembled budding formations after 24 h, becoming more circular after seven days (Figure 4A); some were jellyfish-shaped (Figure 4C).

Second, the IHC analysis of MUG CCArly and the CCTM assessed whether CCTM resembled the in vivo situation. Spheroids differed in composition and structure. MUG CCArly spheroids were structured less tightly than CCTM spheroids. MUG CCArly cells appeared to form circles and tube-like structures (red arrows) that were embedded in a matrix compound (Figure 4B, left column). These MUG CCArly spheroids stained positive for the CCA marker CK7 and negative for the stromal marker vimentin. In CCTM spheroids, CCArly CAF stained positive for vimentin. CCArly CAFs embedded MUG CCArly tumor cells in a tube-like formation in the middle of spheroids (red arrows). Tumor cells were also found at spheroid edges covering the whole surface as a tumor cell monolayer (Figure 4B, right column).

A fluorescence assay, staining viable and dead cells, was performed (Figure 4C) to ensure cell viability in the 3D tumor model. Viable single- and co-cultured spheroids were brightly stained with Calcein AM, whereas the red ethidium homodimer-I dye, indicating dead cells, could be detected toward the margin of CCTM spheroids (Figure 4C) and in CCArly CAF spheroids (Appendix A). Merging all three dyes, viable spheroids illustrated by green colors were confirmed for all culturing conditions. Dead spheroids depicted as violet colors in merged images could only be observed for dead control spheroids that were previously treated with 0.2% Triton X-100 (Figure 4C).

### 3.4. Mass Spectrometric Analysis of 3D Cultures

Since spheroid viability could be verified until day nine, two time points were selected for proteomics. Spheroids were analyzed for proteomic differences between MUG CCArly single-culture and co-culture spheroids by LS-MS/MS. Protein concentration measurements of all three groups revealed the highest values in the co-culture tumor model (Appendix A). Two time points, at day three and seven, were chosen to demonstrate changes over time, as visualized by a principal component analysis (PCA). Biological replicates of single-culture spheroids of the MUG CCArly and CCArly CAF group clustered tightly at both time points (Appendix A). In contrast, the co-culture tumor model showed individual clusters for the measured time points. Comparing both time points, 996 significantly differently expressed proteins were detected in MUG CCArly spheroids, 478in the CCArly CAF group, and the majority of 2412 proteinsin the CCTM group (Appendix A).

### 3.5. The Angiogenic Proteome Is Enriched Particularly in the CCTM

Overall, 7484 proteins were detected and are visualized in heat maps in Figure 5A for day three and Figure 6A for day seven (Appendix A).

Comparing MUG CCArly single-culture spheroids with the CCTM model, 1087 significantly differently expressed proteins with a fold-change expression above 2 and below 0.5 were observed on day three. Of these, 818 proteins were overexpressed in the CCTM group and 228 in MUG CCArly spheroids (Figure 5B). These differentially expressed proteins were reduced nearly by half for day seven, with 594 overexpressed proteins in the CCTM, and 80 in the MUG CCArly group (Figure 6B).

In the MUG CCArly group on day three, the cellular components were mainly enriched in GO terms related to ‘vesicles’ and the ‘extracellular space’. Considering biological functions, 6.98% could be assigned to processes connected with ‘angiogenesis’, ‘blood vessel morphogenesis’, and ‘vasculature development’ (Appendix A). In CCTM spheroids, almost half of the enriched proteins correlated with metabolic processes (50.72%); followed by 9.54% with biological processes related to ‘extracellular matrix organization’ including glycosaminogylcan, aminoglycan, and proteoglycan metabolic processes; 19.76% with ‘immune system processes’; and 10.48% with ‘angiogenesis’, ‘blood vessel morphogenesis’, and ‘vasculature development’. At day seven, this ratio changed in CCTM spheroids to 53.45%, 12%, 19.45%, and 10.36%, respectively.

Scrutinizing angiogenesis-related proteins to compare MUG CCArly single- and co-cultured spheroids and considering only proteins above a 2-fold and below a 0.5-fold change, 78 angiogenesis-related proteins were detected in the CCTM and 18 in the MUG CCArly group on day three, showing protein–protein interactions as well (Figure 5C,D).

Seven of these proteins were enriched when comparing the CCTM to both single cultures as summarized in Table 1.

SerpinE1 with a 187-fold changed expression compared to MUG CCArly and a 2.9-fold expression compared to CCArly CAF on day three are on top of this ranking, followed by PTGS2 with a 172-fold and 5.7-fold expression, respectively. Two genes that were highly enriched in both group comparisons were CXCL8, expressing the protein IL8, and the leukemia inhibitory factor (LIF) (Table 1). Fold-change expressions decreased for all genes, especially when comparing MUG CCArly and CCTM until day seven. CXCL8 and LIF, but also IL6 were found to be enriched in the CCTM group for both single-culture comparisons (Table 1). Raw data for IL6 at day three revealed that two out of five biological replicates of MUG CCArly were below the detection limit (NaN), leading to an exclusion of IL6 for further analyses. Comparison of the remaining three measured replicates to the CCTM group showed the same trend.

### 3.6. CCArly CAF Contribute to An Angiogenic TME

To address proteins that were automatically excluded from a differential expression analysis in group comparisons, important secreted proteins that were related to angiogenesis were measured by Luminex xMAP technology. This method yielded concentrations that were normalized to the total spheroid protein concentrations. TGF-beta could not be detected for all three conditions. MUG CCArly spheroids expressed highest PDGFB levels (Figure 7K), whereas most of the measured analytes were very low or beneath the detection limit, for MUG CCArly spheroids and included IL6, VEGF-A, IL8, HGF, MMP2, SDF1, PAI1 Tenascin C, FGF-2, and VEGF-D (Figure 7A–I,L). MCP1 was expressed equally in all three groups on day three, with an even higher expression in MUG CCArly on day seven (Figure 7J). Compared to MUG CCArly 3D cultures, the highest expression in the CCTM group was observed for the pivotal angiogenic factor VEGF-A, with big differences on day seven, also versus CCArly CAF (Figure 7B). Two further analytes, IL6 and IL8, displayed highest levels in the CCTM group, closely followed by CCArly CAF single-culture spheroids (Figure 7A,C). CCArly CAF and CCTM spheroid IL6 secretion exceeded highest assay standards. However, considering raw data, highest fluorescence intensities were detected in the CCTM group. Comparing CCArly CAF single cultures to CCTM spheroids, highest expression differences were seen for MMP2, SDF1, Tenascin C, and, most notably, HGF (Figure 7D–F,H,). Overall, all ten angiogenic factors that were released could be assigned to CCArly CAF.

## 4. Discussion

CCA is an aggressive and rare bile duct cancer with a low overall survival rate and increasing incidences worldwide. The most common type is pCCA, at 50–60%. Only a few cell lines of this origin are available for research [28,29] and disease pathophysiology remains obscure. CCA-specific TME typically consists of a high proportion of CAFs, which contributes to tumor progression, angiogenesis, metastasis, and therapy resistance [30,31,32,33]. In order to map the central role of CAFs in CCA, a new autologous human 3D co-culture tumor model, the CCTM, was developed here.

A stable expression of CCA tumor cell markers was maintained throughout the in vitro and in vivo experiments, reflecting an IHC profile comparable to other available CCA cell lines [11]. MUG CCArly presents tumorigenicity, whereas the combination with CCArly CAF increases the tumorigenic potential. These observations corroborate a study in which primary non-autologous CAF co-injected with the HuCCT-1 CCA cell line promoted tumor growth in vivo [10]. Additionally, the non-tumorigenic potential of CCArly CAF alone confirms previous findings [34]. Even in the combination group, CCArly CAF could not be detected, as demonstrated by negative staining for human vimentin.

The in vivo context can be better simulated in a 3D in vitro multicellular spheroid culture than in 2D cultures, resembling the ECM-microenvironment, gene expression profile, cell-to-cell signaling, and tumor promoting properties more closely [17]. For example, Ni, Makino, and Tabata conducted experiments with CAFs and their migratory influence on small-cell lung cancer cells [35]. Another study on non-small-cell lung cancer underlines the influence of TME in an in vitro tumor multicellular spheroid model on normal lung fibroblast and monocyte cell lines, which differentiated into CAF and tumor-associated macrophages (TAM), respectively [36]. Work by Alzeeb et al. exemplifies the use of primary CAFs and their varying influences on two human gastric cancer cell lines co-cultured in a direct 3D model [37].

In contrast to these studies, our aim was the establishment of a 3D tumor model using stromal and tumor cells originating from the same patient material, as already depicted by other researchers. An example was illustrated by Gao et al., using primary patient material and describing the pivotal role of CAFs on the invasive behavior of serous ovarian cancer (SOC), distinguishing high-grade SOC in presence of CAFs from low-grade cases [38]. All these studies highlight the significance of stroma–tumor interactions and strongly encouraged us to generate a 3D multicellular spheroid model for the rare CCA. Furthermore, we decided to use CAFs and corresponding tumor cells from the same patient suffering from the most common form of CCA, the pCCA, but with the fewest cell lines available for research. We wanted to select a reproducible method that could be used by other researchers utilizing standard laboratory materials and techniques. An additional important feature in defining the method was to obtain spheroids that could be easily reused in further 3D assays. Therefore, we decided on the cultivation in a scaffold-free and anchorage-independent approach using ULA plates as the most suitable method.

With this focus on 3D cell cultures aimed at closely mimicking these desmoplastic interactions, 3D single- and co-cultures of both cell lines were generated, with MUG CCArly alone not being able to build spheroids. However, using collagen I, compact 3D structures were formed, which supports previous results using various media supplementations [39]. Comparing MUG CCArly single- and co-cultures, the CCTM structures resembled the primary tumor tissue; in 3D single cultures, MUG CCArly formed less densely packed structures embedded in a matrix, probably consisting of the supplemented collagen I. Reportedly, collagen I expressed by CAFs generated stiffness without promoting CCA growth [40]. As a matrix component of CCA TME, collagen I had also been used as a gel matrix for an organotypic rat CCA in vitro model in another study [41]. Proteomics identified collagen I, composed of *COL1A1* and *COL1A2*, in CCArly CAF and CCTM spheroids and underlined its suitability for spheroid supplementation.

Life/dead staining of the 3D model illustrated a weak ethidium-homodimer-I (EthD-I) staining for CCArly CAF spheroids on day nine, despite a high viability illustrated by Calcein AM. EthD-1 fluorescence can be detected when bound to dsDNA, RNA, and triplex nucleic acid structures, but also oligonucleotides. Since in our proteomics approach, no evidence to cell death could be observed, not even by the proteomics approach, other factors, such as the recently detected glycol-RNAs presented on cell surfaces, could be accountable. Some of these non-coding glycol-RNAs were shown to interact with Signlec immune suppressors presented on cell surfaces on cancer cells and monocytic cells, indicating a possible role in immune evasion [42,43,44]. However, our aim was the direct comparison of single- and co-cultivated MUG CCArly cells and these spheroids illustrated a highly viable phenotype; we did not further investigate this aspect.

Proteomic analyses comparing single- and co-cultures of MUG CCArly, including CCArly CAF, illustrated the importance of CAF in the CCA TME. CCArly CAF displayed the smallest protein composition changes over time and a stable growth behavior of the cells. The biggest differences in protein changes and the highest total protein concentration were observed in the CCTM group, suggesting a higher metabolic activity under 3D co-cultivation conditions compared to single 3D cultures.

A decrease in proliferation-related proteins was noted in CCTM spheroids over time, while the GO terms that related to metabolic processes, extracellular matrix remodeling, immune system processes, and angiogenesis were upregulated. Angiogenesis-related proteins were mainly detected in the CCTM group, but not in MUG CCArly spheroids. Angiogenesis involves the reorganization of ECM components, including a local degradation of ECM for neovascularization. Along with an inflammatory microenvironment, tumor neovascularization for angiogenesis and lymphangiogenesis contributes to invasiveness and metastasis formation [30,45,46].

These processes are consistent with the published literature and they indicate that CCTM simulated the physiological in vivo state of CCA more closely than the CCA single culture [47]. Over 70 angiogenesis-related proteins were detected on day three, decreasing until day seven, which may be attributed to the stringent threshold that included only proteins above a two-fold change. Additionally, only proteins significantly enriched in the CCTM group compared to both single-culture conditions were listed, and again, only proteins above the two-fold threshold and those detected at both measured time points were included. All other factors below the detection limit were automatically excluded from proteomic analysis.

Inflammation in CCA often leads to carcinogenesis with persistent IL6 secretion promoting tumor cell survival and growth [48,49,50]. High levels of the inflammatory cytokines, IL6 and IL8, were detected only in the CCTM and by CCArly CAF. Both factors have been implicated in angiogenesis and were detected in protein profiling and supernatants. IL8 and IL6 protein expression increased as a function of time, indicating metabolically active spheroids. IL8 has been shown to correlate with advanced TNM staging in pCCA [51]. This multifunctional factor is also associated with modifications in immune responses, chemotherapy resistance, tumor progression, and metastasis [51,52,53,54,55,56]. Cyclooxygenase 2 (COX-2) expression has also been associated with TNM staging where COX-2 inhibition decreased the migratory behavior of CCA cells [57]. Increased COX-2 levels, found especially in the CCTM group, suggest an increased migratory behavior.

The angiogenic, immunomodulatory and chemoattractant factor MCP-1 was highest in day seven MUG CCArly cultures compared to CCArly CAF and the CCTM, and is associated with infiltrating monocytes that become tumor-associated macrophages (TAM). At the invasive tumor front, polarized M2 macrophages often occur while contributing to chemoresistance, ECM changes, immunosuppression, and angiogenesis [58,59].

Here, other prominent factors such as TNC expression are linked to MMP upregulation, angiogenesis, and further metastasis formation [60]. CCArly CAFs were the main source of secreted TNC and MMP2. Increased MMP2 and TGFBI levels had previously been associated with poor prognosis in CCA [61,62].

Additionally, ECM-1, the aforementioned IL8, PAI-1, SDF-1, and HGF, are angiogenesis-related proteins that are previously linked to the invasive behavior of tumor cells [63,64,65,66]. Soluble SDF-1 as well as HGF decreased for day seven CCTM cultures, whereas no significant difference could be detected via the proteomics approach comparing CCArly CAF to CCTM. Endocytosis of these factors was previously described and could reflect the direct crosstalk between soluble factors of CCArly CAFs with MUG CCArly cells [67,68,69,70,71].

The synergistic, angiogenic potential of HGF and VEGF was reported in vivo and in vitro [72]. VEGF-A is a growth factor with the highest relevance for angiogenesis and a strong lymphangiogenic potential [73]. On day seven, the highest angiogenic potential was detected using VEGF-A levels secreted by CCTM spheroids, indicating an increased angiogenic potential in co-cultures. In proteomics, VEGF-A was automatically excluded from differential expression analysis since VEGF-A levels were below the detection limit in the MUG CCArly spheroids that were used as comparison. Neuropilin-2 (NRP2), a co-receptor of VEGF-A associated with angiogenesis, survival, and therapy resistance [74], was slightly elevated in day three CCTM cultures but was below the two-fold change threshold on day seven. Chemoresistance, a typical feature of CCA, was also shown to be induced by the leukemia inhibitory factor (LIF) through the PI3K/AKT pathway [75]. Furthermore, increased LIF levels in a murine mesenchymal stem cell study correlated with upregulated angiogenic genes coding for IL8, MCP1, and VEGF [76], which the present findings confirm.

A complete replacement of in vivo experiments can hardly be achieved by an in vitro model, since many factors such as an immune and blood-circulation system are missing. Our CCTM could aid researchers in basic research to investigate the crucial role of CAFs on CCA tumor progression. This knowledge might help find new hypotheses concerning how to treat this rare cancer, also taking CAFs into account. Three-dimensional cell cultures are already used to define treatment options via high-throughput screening. For more complex studies, our CCTM could be improved by the incorporation of further cell types [77,78,79].

## 5. Conclusions

In summary, the presented proteomic and multiplex data illustrate the high angiogenic potential under the 3D CCTM conditions and stress the importance of including CAFs in a CCA tumor model.

The novel CCTM in vitro model is a reproducible, well-characterized, and easy-to-handle model that mimics the physiology of CCA. It is time-efficient to generate compared to existing methods such as freshly isolating primary cells or differentiating-induced pluripotent stem cells. Researchers using our cell lines will take a maximum of one month to recreate our day seven CCTM model under standard laboratory conditions.

We are planning to use our CCTM to further investigate tumor desmoplasia, signal transduction, tumor progression, and cell-to-cell communication, as well as use it for therapy and resistance studies in vitro and in vivo.

Future research should further investigate the angiogenic potential of the presented CCTM model and include additional cell types to extend the TME. Investigations could address tube formation by endothelial cells or the polarization of monocytes into tumor macrophages and their contribution to the invasive nature of cancer.

## Figures and Tables

**Figure 1 cancers-15-01757-f001:**
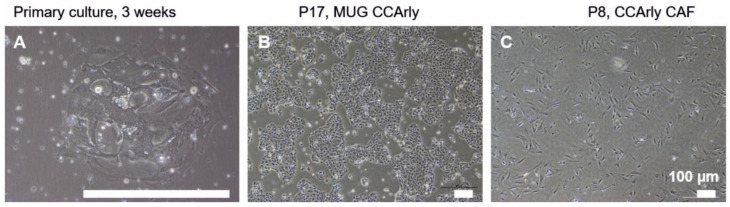
Primary cell culture of MUG CCArly and CCArly CAFs over time. (**A**): Primary CCA culture at p0; (**B**): continuous cell line MUG CCArly, p17; and (**C**): CCArly CAF, p8. Scale bar: 100 μm.

**Figure 2 cancers-15-01757-f002:**
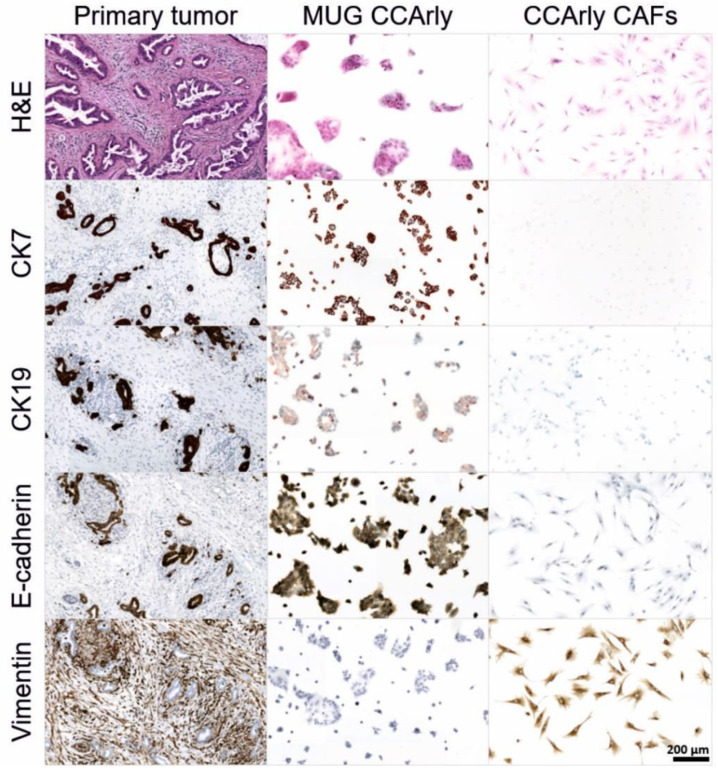
CCA markers expressed in the primary tumor tissue compared to both continuous cell lines MUG CCArly and CCArly CAF. Scale bar: 200 μm.

**Figure 3 cancers-15-01757-f003:**
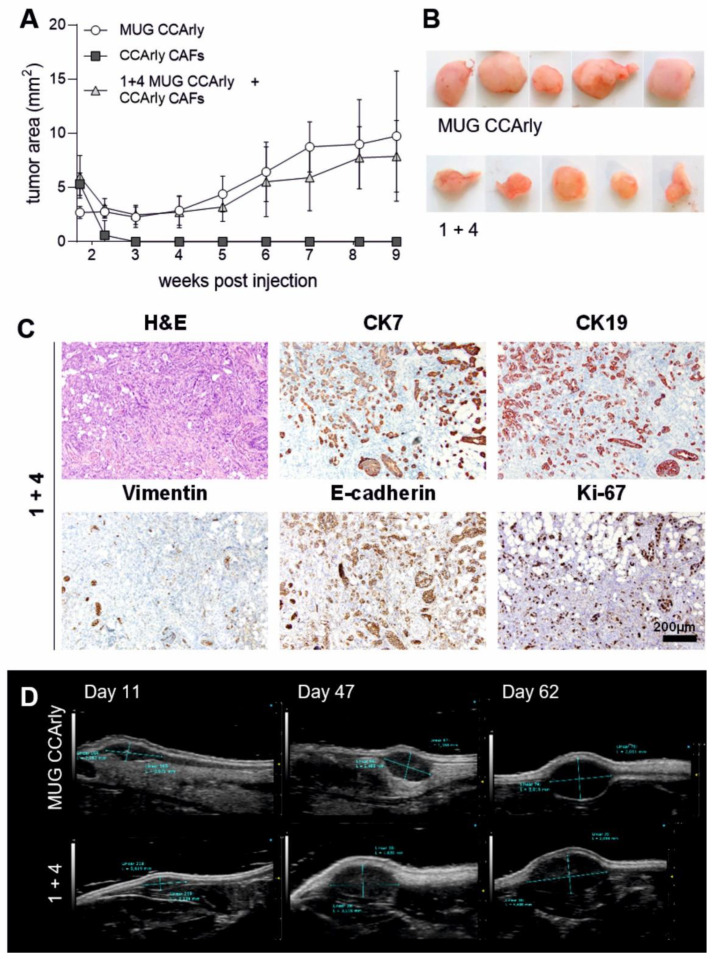
Tumorigenic potential of MUG CCArly and CCArly CAFs in nude mice. (**A**): Tumor growth over time for all three groups, as well as dissected xenografts of both groups (**B**) are shown. (**C**): IHC staining for CCA markers CK7, CK19, and e-cadherin, as well as vimentin for human stromal cells and the proliferation marker Ki-67 are illustrated for the 1 + 4 combination of MUG CCArly with CCArly CAFs, respectively. (**D**): High frequency ultrasound (HF-US) images of one representative mouse for each group are illustrated. Scale bar: 200 μm; 1 + 4: combination group injected with 1 part MUG CCArly and 4 parts CCArly CAF.

**Figure 4 cancers-15-01757-f004:**
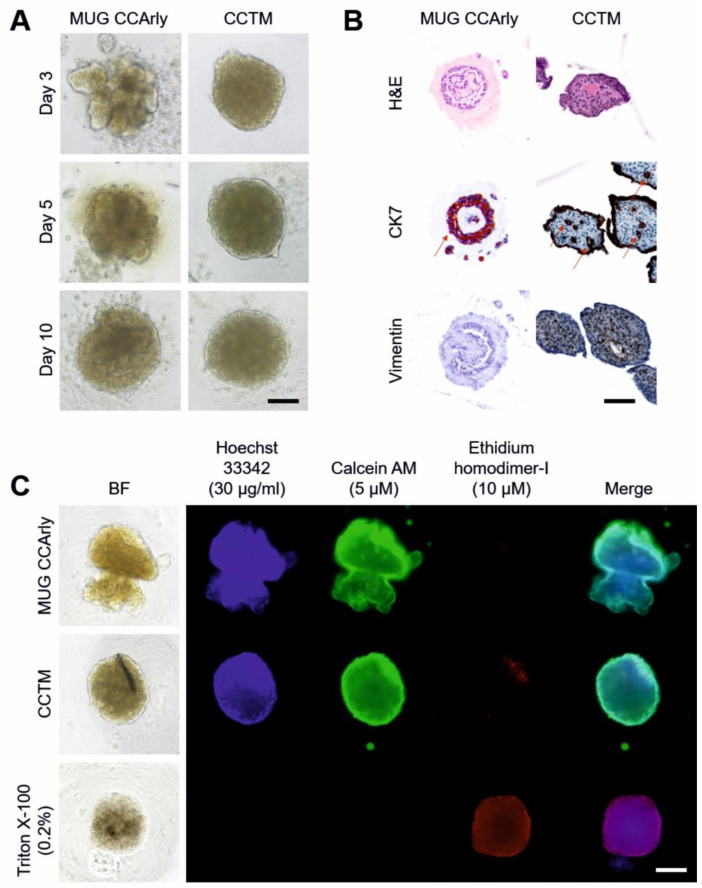
Quality check of single- and co-cultured spheroids MUG CCArly. (**A**): Spheroid formation over time under the bright field microscope, and MUG CCArly cultures were supplemented with 28 μg/mL bovine collagen I. (**B**): IHC of MUG CCArly and the CCTM for the CCA marker CK7 and the stromal marker vimentin. Red arrows indicate tube-like formations. (**C**): Live/dead fluorescently stained spheroids on day nine. Single-cultured spheroids and the CCTM were compared to dead control spheroids, permeabilized by 0.2% Triton X-100. Hoechst 33342, cell nuclei; Calcein AM, viable cells; Ethidium homodimer-I, dead cells; merge, all three dyes merged. CCTM: CCArly co-culture tumor model (1 + 4 ratio); BF, bright field; and scale bar, 100 μm.

**Figure 5 cancers-15-01757-f005:**
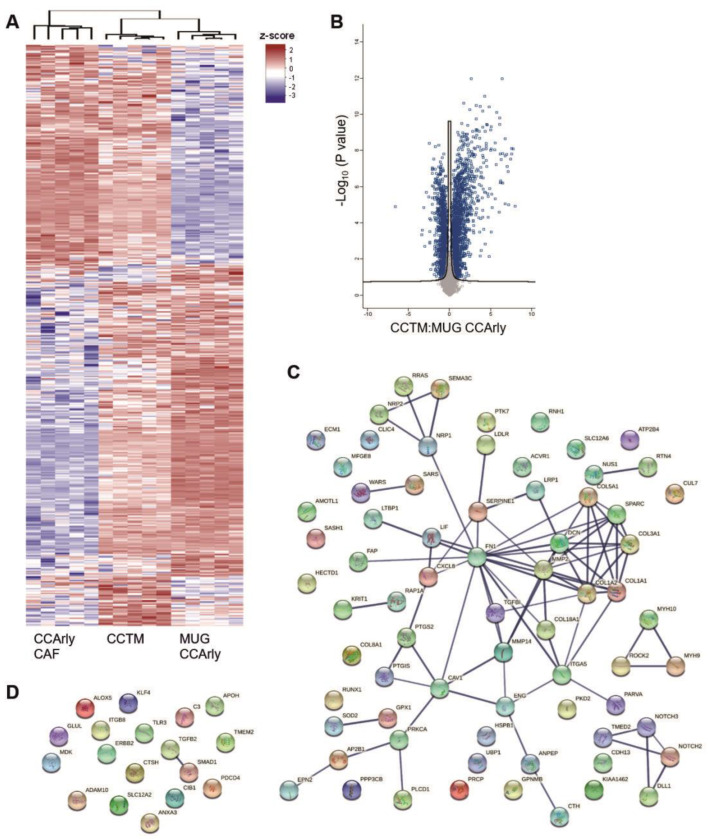
Proteome of single- and co-cultured MUG CCArly spheroids of day 3. (**A**): Heat maps illustrating the total amount of significantly different identified proteins. (**B**): Volcano plot comparing proteins of MUG CCArly and CCTM spheroids. Blue colored rectangles illustrate significant proteins including *p*-value correction. Angiogenic proteins overexpressed in CCTM-(**C**) and MUG CCAry spheroids (**D**) with a fold-change value above 2 each. FDR = 1%. *n* = 5. CCTM: MUG CCArly co-culture tumor model.

**Figure 6 cancers-15-01757-f006:**
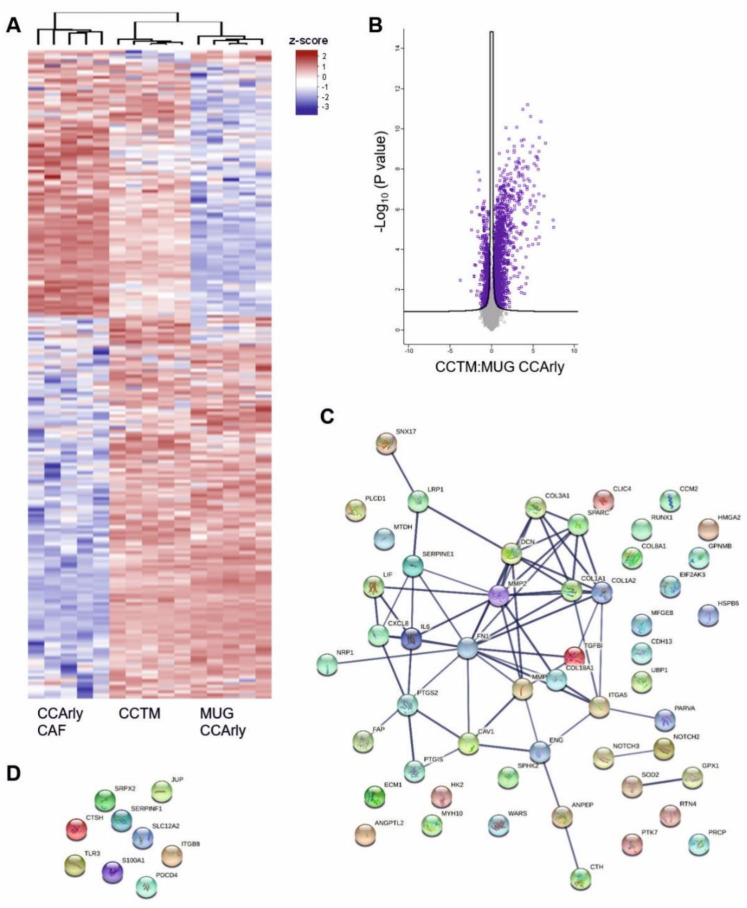
Proteome of single- and co-cultured MUG CCArly spheroids of day 7. (**A**): Heat maps illustrating the total amount of significantly different identified proteins. (**B**): Volcano plot comparing proteins of MUG CCArly and CCTM spheroids. Purple colored rectangles illustrate significant proteins including p-value correction. Angiogenic proteins overexpressed in CCTM-(**C**) and MUG CCAry spheroids (**D**) with a fold-change value above 2 each. FDR = 1%. *n* = 5. CCTM: MUG CCArly co-culture tumor model.

**Figure 7 cancers-15-01757-f007:**
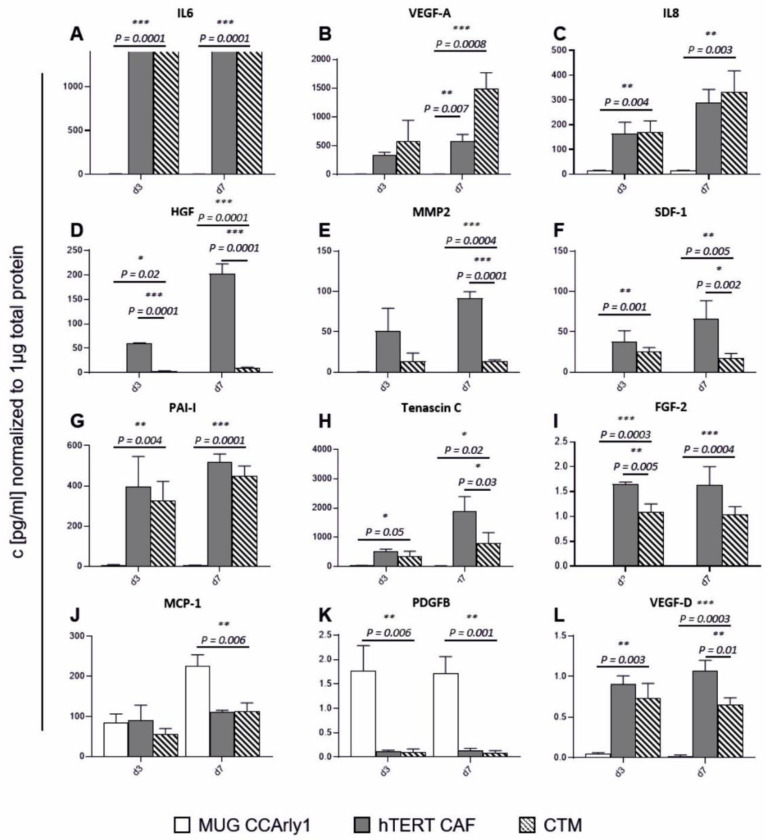
Soluble angiogenic factors expressed in MUG CCArly single- and co-cultured spheroids with CCArly CAF. Following factors are illustrated: IL6 (**A**), VEGF-A (**B**), IL8 (**C**), HGF (**D**), MMP2 (**E**), SDF-1 (**F**), PAI-1 (**G**), Tenascin C (**H**), FGF-2 (**I**), MCP-1 (**J**), PDFGB (**K**) and VEGF-D (**L**). *n* = 3, *p* ≤ 0.05 = *, *p* ≤ 0.01 = **, and *p* ≤ 0.001 = ***.

**Table 1 cancers-15-01757-t001:** Fold-change ratios of day three and day seven angiogenic proteins enriched in the CCTM group, shared within both single-culture groups, either with MUG CCArly or CCArly CAF spheroids. Fold change > 2 in both group comparisons, *p* < 0.05. IL6 was below the detection threshold in MUG CCArly single cultures.

			Day Three		Day Seven	
			Fold Change			
Protein ID	Protein Name	Gene	CCTM: MUG CCArly	CCTM: CCArly CAF	CCTM: MUG CCArly	CCTM: CCArly CAF
P05121	PAI1	*SERPINE1*	187.4341	2.9192	30.8141	2.2039
P35354	COX-2	*PTGS2*	172.0948	5.7055	42.3465	5.7281
P10145	IL8	*CXCL8*	109.7289	26.4974	30.8723	18.0813
P15018	LIF	*LIF*	71.3025	41.38	19.1683	12.0751
Q15582	BGH3	*TGFBI*	22.3808	3.1987	19.4534	2.7792
Q16610	ECM1	*ECM1*	9.7985	2.6402	2.8068	2.6888
O60462	NRP2	*NRP2*	2.179	2.2517	<2	2.5282
P05231	IL6	*IL6*	-	-	47.3197	25.7217

## Data Availability

Data are available from the corresponding author upon reasonable request.

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
