# Peer review of "MUG CCArly: A Novel Autologous 3D Cholangiocarcinoma Model Presents an Increased Angiogenic Potential"

_cancers, 2023, doi:10.3390/cancers15061757_

Round 1

Reviewer 1 Report (Previous Reviewer 1)

I recommend the publication.

Reviewer 2 Report (Previous Reviewer 3)

The authors have nicely addressed all of the comments.

Reviewer 3 Report (Previous Reviewer 2)

The authors replied to my comments accordingly.

This manuscript is a resubmission of an earlier submission. The following is a list of the peer review reports and author responses from that submission.

Round 1

Reviewer 1 Report

This research is interesting, but the novelty is not clear. For example, the section Introduction is too poor to understand the concept of 3D models using CAF. Taken together, major revision should be made before re-submission. This manuscript would be re-considered only when all the comments were responded to correctly.

1. Introduction or Discussion

The authors should mention the concept of the 3D cancer-CAF models and discuss the novelty of this study by comparing these references.

Review (for concept)

Cancers 202012(10), 2754

https://doi.org/10.1002/bit.26845

Research papers

J Exp Med (2019) 216 (3): 688–703.

Tissue Eng. Part C Methods 201925, 711–720. https://doi.org/10.1089/ten.tec.2019.0189

J. Vis. Exp. (156), e60660

2. figure 7

Statistical analysis should be performed.

Author Response

Thank you for your valuable review of our manuscript please find our point by point answers in the attached file

Reviewer 2 Report

In their manuscript ´MUG CCArly: a novel autologous 3D cholangiocarcinoma model presents an increased angiogenic potential´, Schrom et al. successfully isolated autologous tumor cells and tumor-associated fibroblasts from a patient and characterized the isolated cells alone or in co-culture in detail regarding their angiogenic potential. The data provide an important tool for studying tumor-stromal cross-talk, angiogenesis, invasion and are suited for drug or drug resistance testing. There are few comments that the authors may consider to include.

1. Why are the tumor pieces cultured for one month prior separating tumor from stromal cells? What do the authors mean with different detachment points? Are the cells kept in culture for 8 months prior pure CCA cultures are visible (line 256, Fig. 1B)?

2. Why were A549 cells used as controls?

3. Could the tumorigenic potential of MUG CCArly be assessed by in vitro methods? Can CCTM replace in vivo experimentation? If so, the authors could discuss this point.

4. Could the authors display an improved Fig. 4, since the ethidium homodimer staining is very weak even in the TritonX-100 control?

5. Line 252: remove This or A from ´This A primary Klatskin tumor tissue...´

Author Response

We thank you for your valuable review of our manuscript. You will find our response point by point in the attached file

Reviewer 3 Report

I thank you authors for the opportunity to review their manuscript regarding the in-vitro and in-vivo characterization of a new cholangiocarcinoma (CC) cell line and associated CAF cell line. 

Overall, the manuscript is well written.  Relative to other cancers, there are fewer well established CC cell lines and I commend the researchers for their work.

Specific comments:

In this manuscript, the authors isolated CC cells and stromal cells from a single CC patient. Assuming that the patient consented to the establishment of this cell line - It would be recommended to add that to the methods.

3.1

Figure 1. Are all the images from the same magnification? image 1A appears to be from a higher magnification.

Figure 2.  are there any markers specific to CAFs, and not just stromal cells (such as FSP-1, PDGFR, SMA alpha, CD36)? 

Did the authors try to compare CCarly CAF to using just other available immortalized fibroblasts?

3.2

Figure 3.C - were any measures of angiogenesis assessed from the in-vivo specimens? vascularity on H&E, staining for VEGF, CD34, CD31, CD105 ?  

Since desmoplasia is a key feature of the CC TME and the authors are aiming to show that the model recapitulates the actual CC TME - were there any stainings for Trichrome, alpha SMA, Desmin?

3.3

The supplementary figure S2 of the CCarly CAF spheroids is oddly positive for the Ethidium homodimer-I stain (as positive as the TritonX control in Figure 4C.

3.5

Figures 5B and 6B -would recommend using the word "Volcano" instead of "Vulcano"

3.6

Figure 7. Please add "*" to denote statistical significance. have the authors assessed PDGF or TGFb? it is interesting that aside from VEGF on day 7, it seems as though co-culturing resulted in a decrease in expression of HGF and SDF-1. What do the authors make of this?

Conclusions

The conclusions are fully supported by the data:

A. it is still not clear whether CCarly CAFs are truly CAFs (from a biomarker expression standpoint)

B. Time-efficient - the authors describe an eight-months process for creation of these cells.

C. "Can be used to assess desmoplasia, signal tranduction studies, tumor progression and cell-to-cell communication" - these all remain to be studied, as neither was assessed in this study we do not have any data to support that this model is appropriate for these studies. The authors may consider revising their statement to suggests assessing these features in future studies.

Author Response

(The authors gave the same response as above.)
